# Reverse Shoulder Arthroplasty with Bony and Metallic versus Standard Bony Reconstruction for Severe Glenoid Bone Loss. A Retrospective Comparative Cohort Study

**DOI:** 10.3390/jcm10225274

**Published:** 2021-11-13

**Authors:** Marko Nabergoj, Lionel Neyton, Hugo Bothorel, Sean W. L. Ho, Sidi Wang, Xue Ling Chong, Alexandre Lädermann

**Affiliations:** 1Valdoltra Orthopaedic Hospital, 6280 Ankaran, Slovenia; mmarkoj@gmail.com; 2Faculty of Medicine, University of Ljubljana, 1000 Ljubljana, Slovenia; 3Ramsay Générale de Santé, Hôpital Privé Jean Mermoz, Centre Orthopédique Santy, 69008 Lyon, France; neyton.lionel@gmail.com; 4Research Department, La Tour Hospital, 1217 Meyrin, Switzerland; hugo.bothorel@latour.ch; 5Department of Orthopaedic Surgery, Tan Tock Seng Hospital, Singapore 308433, Singapore; Sean_WL_HO@ttsh.com.sg; 6Division of Orthopaedics and Trauma Surgery, La Tour Hospital, 1217 Meyrin, Switzerland; sidi99.wang@gmail.com (S.W.); chongxueling@gmail.com (X.L.C.); 7Faculty of Medicine, University of Geneva, 1211 Geneva 4, Switzerland; 8Division of Orthopaedics and Trauma Surgery, Department of Surgery, Geneva University Hospitals, 1205 Geneva, Switzerland

**Keywords:** shoulder, prosthesis, defect, reconstruction, autologous graft, survivorship, loosening, integration

## Abstract

There are different techniques to address severe glenoid erosion during reverse shoulder arthroplasty (RSA). This study assessed the clinical and radiological outcomes of RSA with combined bony and metallic augment (BMA) glenoid reconstruction compared to bony augmentation (BA) alone. A review of patients who underwent RSA with severe glenoid bone loss requiring reconstruction from January 2017 to January 2019 was performed. Patients were divided into two groups: BMA versus BA alone. Clinical outcome measurements included two years postoperative ROM, Constant score, subjective shoulder value (SSV), and the American Shoulder and Elbow Surgeons Shoulder (ASES) score. Radiological outcomes included radiographic evidence of scapular complications and graft incorporation. The BMA group had significantly different glenoid morphology (*p* < 0.001) and greater bone loss thickness than the BA group (16.3 ± 3.8 mm vs. 12.0 ± 0.0 mm, *p* = 0.020). Both groups had significantly improved ROM (anterior forward flexion and external rotation) and clinical scores (Constant, SSV and ASES scores) at 2 years. Greater improvement was observed in the BMA group in terms of anterior forward flexion (86.3° ± 27.9° vs. 43.8° ± 25.6°, *p* = 0.013) and Constant score (56.6 ± 10.1 vs. 38.3 ± 16.7, *p* = 0.021). The BA group demonstrated greater functional and clinical improvements with higher postoperative active external rotation and ASES results (active external rotation, 49.4° ± 17.0° vs. 29.4° ± 14.7°, *p* = 0.017; ASES, 89.1 ± 11.3 vs. 76.8 ± 11.0, *p* = 0.045). The combination use of bone graft and metallic augments in severe glenoid bone loss during RSA is safe and effective and can be considered in cases of severe glenoid bone loss where bone graft alone may be insufficient.

## 1. Introduction

Addressing severe glenoid deficiency during shoulder arthroplasty is technically challenging. Glenoid deficiencies have been reported in up to 39% of patients undergoing reverse shoulder arthroplasty (RSA). Such glenoid bone loss may occur in any part of the glenoid, including the posterior aspect (18%), superior aspect (9%), anterior aspect (4%), or as a global erosion (6%) in patients undergoing RSA [1,2].

Implantation of RSA in patients with advanced deformity of the glenoid may lead to several problems due to malpositioning of the glenoid baseplate. Excessive medialization of the glenoid baseplate causes muscle shortening with decreased tension resulting in poorer function [3,4]. Diminished deltoid wrapping around the greater tuberosity can also increase the risk of prosthetic instability and cosmetic deformity [5]. Additionally, the excessive medialization results in increased scapular notching with inferomedial glenoid bone erosion and polyethylene wear [6]. In superior glenoid bone loss, there is a risk of placing the glenoid baseplate in superior inclination. This has been shown to be an important risk factor for aseptic loosening as it increases shear forces and decreases compressive forces that otherwise stabilize RSA [7,8].

To avoid these negative outcomes, surgeons often attempt to reconstruct the glenoid bone loss, allowing for an optimal positioning of the baseplate. The common approach is to use the humeral head autograft to fill glenoid defects. However, there are some technical considerations. Firstly, the amount of humeral head autograft available is not always sufficient to fully compensate for the bone defect. Secondly, it should be noted that in such complicated glenoid reconstruction, graft incorporation requires stabilization through a peg inserted in a native glenoid [9]. These factors add to the technical difficulties. One alternative is to use a metallic augment to compensate for the glenoid bone loss [10]. However, in severe glenoid deficiencies, the available metallic augments may not be adequately thick enough to fully reconstruct the glenoid bone loss. In these cases, a combination of both metallic augment and bone graft can be utilized to sufficiently build up the bone loss. To our knowledge, no study has evaluated the combination of an augmented baseplate with bone grafting for RSA with severe glenoid defects.

The aim of this study was to assess the clinical and radiological outcomes of a combined bony and metallic augmented baseplate for RSA with severe glenoid defects. The hypothesis was that the combined use of bone graft and metallic augments in severe glenoid bone loss during RSA is safe and effective.

## 2. Materials and Methods

### 2.1. Study Design, Data Collection, and Ethical Committee Approval

Between January 2017 and January 2019, all patients who had an RSA by either a combination of bony and metallic augments or bony augments alone were considered potentially eligible for inclusion in this retrospective analysis of data prospectively collected during the SHOUT (Shoulder OUTcome) multi-center study. The inclusion criteria were a severe glenoid defect, defined by a need to use a graft thicker than 1 cm to restore inclination and version at acceptable values (0 degree and <20 degrees, respectively) using a 3D planning software (Blueprint™|Wright Medical Group, Memphis, TN, USA). The exclusion criteria included avascular necrosis of the humeral head, neurological conditions affecting the upper limb, and patients with less than two years follow-up. Two groups of patients were defined: Group 1 were patients who had only bone graft for glenoid reconstruction during RSA, and Group 2 were patients who had a combination of bone graft and metallic augments for glenoid reconstruction during RSA. The study received ethics committee approval from both centers (CCER 14-227 and COS-RGDS-2021-06-009-NEYTON-L). All the patients gave informed consent for participation in this study.

### 2.2. Surgical Technique and Implant Design

All operations were performed by two experienced [10] shoulder surgeons (A.L. and L.N.) who had performed more than 250 RSAs before the study period. A standard deltopectoral approach was used. A humeral head autograft was harvested and prepared to match the size and location of the glenoid defect. The graft was either temporarily fixed to the native glenoid or held by the post during impaction or screw insertion (Figure 1). The only difference between the two techniques was the baseplate: in the bony-metallic augmentation (BMA) group, a 15 degrees full wedge augmented baseplate (Aequalis™ Perform™ Reversed Glenoid|Wright Medical Group, Memphis, TN, USA) was screwed at the edge of the glenoid with the greatest bone loss (Figure 2). In the bony augmentation (BA) group, a 25-mm-long central peg baseplate (Aequalis Reversed II Glenoid™|Wright Medical Group, Memphis, TN, USA) was impacted into the native glenoid (Figure 2). The type of glenosphere (size and eccentricity) depended on the bone defect, the morphology of the patient, and the tension of the soft tissues. In both groups, the same curved, monoblock short-stem system was used (Aequalis™ Ascend Flex™|Wright Medical Group, Memphis, TN, USA). With this onlay device, the placement of the offset tray affects both humerus lateralization and distalization [11,12]. A 145° neck-shaft angle was used in this study, acquired by using a stem inclination of 132.5° combined with an asymmetric 12.5° polyethylene insert. Stems were cemented if rotational stability was not obtained intra-operatively after insertion of a cement restrictor plug.

### 2.3. Postoperative Rehabilitation

All patients followed the same postoperative rehabilitation protocol. Postoperatively, the arm was placed in an abduction pillow sling for six weeks to promote compression instead of shear forces. After six weeks, the immobilization was discontinued, and active ROM was initiated. Activities of daily living were progressed, but strengthening was not specifically recommended [13].

### 2.4. Study Variables

Patient demographics such as age, gender, diagnosis, side of pathology, hand dominance, body mass index, and tobacco use were collected. Clinical outcomes and radiological outcomes were collected as described below.

### 2.5. Clinical Evaluation

The Constant score, SSV, and ASES were used for clinical assessment. These scores were used for their ease of administration and well-validated data [11,12]. All the patients completed all three scores at the preoperative time point and at the final follow-up of two years. For clinical assessment of ROM, a goniometer was used for the active evaluation of anterior forward flexion and rotations. The external rotation was measured with the arm by the side of the body, whereas internal rotation by the highest vertebral spinous process reached by the patient’s extended thumb.

### 2.6. Radiological Assessment

The initial glenoid bone loss was measured on preoperative CT scans, recorded and classified according to Walch et al. classification [14,15]. The standard anteroposterior view in neutral, external and internal rotation, and axillary lateral view were obtained under fluoroscopic control preoperatively and postoperatively. Using Osirix (Pixmeo, Geneva, Switzerland), postoperative radiographs were assessed for bone graft incorporation defined by the absence of lucent lines observed between the humeral bone graft and the native glenoid, inferior notching at the native glenoid, radiolucent lines (around the peg, screws, and humeral stem), and a shift in the position of the components. The severity of the inferior notching was graded according to Sirveaux classification [16]. Glenoid loosening was confirmed following the criteria of Mélis et al. [17], the criteria being the presence of a radiolucent line >2 mm thick.

### 2.7. Statistical Analysis

The Shapiro–Wilk test was used to check the normality of distributions. Descriptive statistics were presented in terms of means, standard deviations (SD) and ranges for continuous variables and percentages for categorical variables. The significance of pre- vs. postoperative differences within each group was determined using the Wilcoxon signed-rank test for non-normally distributed data and the paired Student’s *t*-test for normally distributed data. The significance of differences between groups was determined using the Mann–Whitney U test (Wilcoxon rank sum test) for non-normally distributed quantitative data and using the Student’s unpaired *t*-test for normally distributed data. For categorical data, the significance of differences between groups was determined using the Fisher exact test. Statistical analyses were performed using R version 3.6.2 (R Foundation for Statistical Computing, Vienna, Austria). *p* values < 0.05 were considered statistically significant.

## 3. Results

There was no significant difference in the demographic data of patients in both groups. Patient characteristics showed no significant differences in terms of patient age, gender, BMI, tobacco usage, or affected side. There was also no significant difference in the preoperative surgical variables such as history of prior surgery, primary diagnosis, glenoid inclination, or glenoid version (Table 1). Patients in the BMA group presented a different glenoid morphology (*p* < 0.001) and a greater bone loss thickness than patients in the bony augmentation (BA) group (16.3 ± 3.8 mm vs. 12.0 ± 0.0 mm, *p* = 0.020). All patients in the BA group had a B2 glenoid defect. In the BMA group, one patient had a glenoid type B1, one patient had a glenoid type B3 and two patients each had a glenoid type D, E3, and C. Preoperative radiological data of each patient are summarized in Table 2. Compared to the BA group, BMA patients had a lower preoperative anterior forward flexion (55.0° ± 38.5° vs. 101.3° ± 31.8°, *p* = 0.010) and worse preoperative Constant scores (18.8 ± 7.4 vs. 34.5 ± 11.7, *p* = 0.013). 

Both the BMA and BA groups completed at least two years of follow-up, with a mean follow-up of 28.1 ± 15.0 and 30.7 ± 10.8 months, respectively. At the final follow-up, both the BMA and BA groups significantly improved their ROM (anterior forward flexion and external rotation) and clinical scores (Constant, SSV, and ASES scores) (Table 3). A greater improvement could be observed in the BMA group in terms of anterior forward flexion (86.3° ± 27.9° vs. 43.8° ± 25.6°, *p* = 0.013) and Constant score (56.6 ± 10.1 vs. 38.3 ± 16.7, *p* = 0.021), probably due to their lower preoperative scores compared to BA patients. However, in the absence of significant preoperative differences, the BA group demonstrated greater functional and clinical improvements than BMA patients with higher postoperative active external rotation and ASES results (active external rotation, 49.4° ± 17.0° vs. 29.4° ± 14.7°, *p* = 0.017; ASES, 89.1 ± 11.3 vs. 76.8 ± 11.0, *p* = 0.045).

At two years follow-up, a bony scapular spur and three inferior graft resorptions were noted in the BA group. In the BMA group, a bony scapular spur, two ossifications in the glenohumeral space, and a grade 1 scapular notching were observed.

## 4. Discussion

This study demonstrates that the combination of bone graft and metallic augmentation of the glenoid baseplate is a safe and effective option to treat severe glenoid deformities during RSA, confirming our hypothesis. Both the BMA and BA groups attained significantly better postoperative functional outcomes (SSV, Constant and ASES scores). Both groups also attained significantly better clinical ROM postoperatively (anterior forward flexion and external rotation). There was a greater improvement in the BMA group with regard to anterior forward flexion (86.3° ± 27.9° vs. 43.8° ± 25.6°, *p* = 0.013) and Constant score (56.6 ± 10.1 vs. 38.3 ± 16.7, *p* = 0.021), but that might be attributed to the significant preoperative differences. Simovitch et al. reported on the minimal clinically important difference (MCID) for different shoulder outcome metrics and ROM after shoulder arthroplasty. They noted that the MCID in terms of active external rotation is 3° ± 2° [18]. Werner et al. showed that patients undergoing reverse shoulder arthroplasty due to rotator cuff arthropathy or glenohumeral arthritis experience a clinically important change if they have at least a nine-point improvement in ASES score [19]. These studies further confirm that in both the BA and BMA groups, the MCID was achieved in both active external rotation and ASES scores. 

Current surgical options that address severe bone loss include preferentially using bone grafting (autograft or allograft) or the use of metallic augmented baseplates (wedge compensation or patient matched implant) [20]. Bone graft can be obtained as an autograft from the autologous humeral head [21,22,23,24] and iliac crest [23], or as a femoral head allograft [21,22,25,26]. However, the quantity or the quality of the graft might not be adequate. Furthermore, the price of allografts or patient matched implants may be unaffordable. Jones et al. directly compared bone grafting versus augmented baseplate and reported similar outcomes in both groups. However, they observed a higher complication percentage in the group with bone graft [27]. To the best of our knowledge, combining an augmented baseplate with a bone graft has not been published yet. The present study shows that integrating a metallic compensation into a bony compensation (compared to native glenoid) is a viable option for extreme bone loss cases. This technique has several advantages. It allows the surgeon to compensate for massive bone defects while avoiding excessive donor-site morbidity by not harvesting an additional bone graft from the iliac crest. It can also relativize the auto- or allograft quality and risk of partial integration. Additionally, we also achieve glenoid lateralization with this technique which decreases scapular notching, and increases ROM and soft tissue tension [9,28,29]. Lastly, adding a full wedge baseplate on a graft creates more inferior tilt, which is key to transforming shear forces into compression ones and promoting graft healing (Figure 3) [5,7,30,31].

In patients with severe glenoid bone loss, integration of the graft is a crucial factor. Recent studies analyzing the use of bone graft in RSA described a satisfactory rate of bone graft incorporation [21,32,33,34,35,36]. However, a systematic review by Malahias et al. still reported a rate of radiographic non-union at 5.2% [37]. In addition, it is important to note that despite evidence of radiological union, true integration of the graft is rarely complete [38], as confirmed in the present study. Given this finding, the authors recommend that as much of the bony defect as possible should be covered by the graft in order to maximize the surface of contact and, consequently, the potential of healing.

The use of a central screw baseplate to fix massive grafts is debatable. It is thought that screws do not provide bone ingrowth possibilities like it is the case around a central peg [38]. The minimal central peg length proposed in the literature that should be inserted in the native bone stock to avoid loosening varies between 8 to 10 mm [21,39]. However, we did not observe signs of glenoid loosening or migration when using a central screw after two years, confirming sufficient stability and the biomechanical findings of Bonnevialle et al. [40]. This observation can be explained by the tremendous compression obtained at the insertion of screw devices and the additional inferior tilt provided by the full wedge baseplate.

The restoration of global lateralization is also essential to improve postoperative function. Humeral offset is heavily influenced by prosthetic design. The use of a curved stem, an eccentric reverse tray with a high offset (3.3 mm), and a 145° neck-shaft angle provides around 10 mm of humeral lateralization that also help to balance the glenoid side [4].

In this study, both the BMA and BA groups achieved significantly better clinical and functional outcomes postoperatively. Regarding active external rotation and ASES scores, there was no significant difference between the groups preoperatively, but the BA group achieved significantly better external rotation and ASES scores postoperatively. As such, the authors recommend that isolated bone grafting be performed for glenoid loss during RSA where possible. In cases where the glenoid bone loss appears too severe for bone grafting alone, a combination of bone graft and metallic augments is a safe and effective option for glenoid reconstruction.

### Strengths and Limitations

To our knowledge, this is the first study that explicitly analyzes the success of a combination of bone grafting and metallic baseplate augmentation with RSA in severe glenoid deformities. The surgeries were performed by experienced, shoulder-fellowship-trained surgeons. This study had several limitations. Firstly, the sample size is small and as such, the analysis may be underpowered. Secondly, the sustained long-term improvement of clinical outcomes and ROM of both surgical techniques remains unclear. Thirdly, the assessment of bone graft incorporation could have been more accurate using a CT compared to the radiograph. Lastly, our study is heterogeneous as different types of glenoid bone erosion were present in each group. Future work should assess a larger population with longer follow-up.

## 5. Conclusions

Combining bone graft and metallic augments in severe glenoid bone loss during RSA is safe and effective, resulting in significantly improved clinical outcomes and ROM. In cases of severe glenoid bone loss where bone graft alone may be insufficient, a combination of bone graft and metallic augments should be considered. 

## Figures and Tables

**Figure 1 jcm-10-05274-f001:**
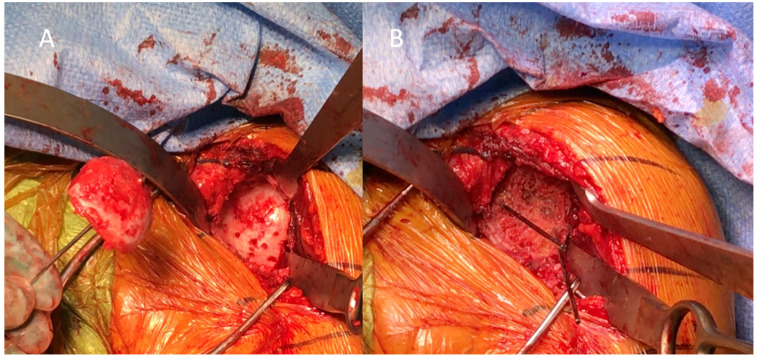
Sagittal view of a left glenoid with severe bone loss. (**A**) The glenoid is prepared with multiple small holes to promote bone healing and graft incorporation. (**B**) The graft is temporarily fixed to the native glenoid before screw insertion. In this case, bone graft alone is able to sufficiently restore the glenoid bone loss.

**Figure 2 jcm-10-05274-f002:**
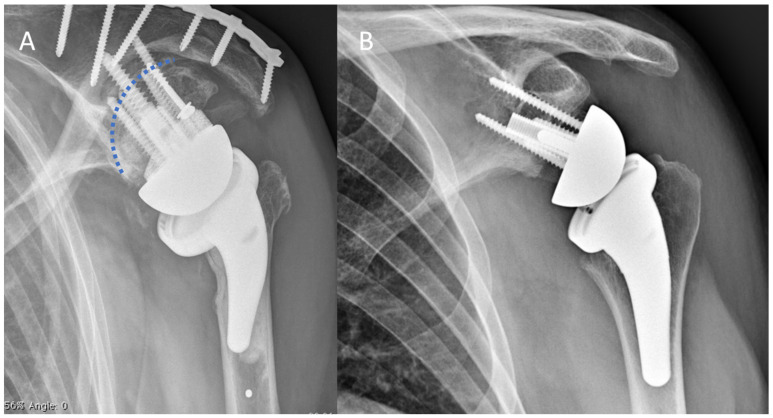
Postoperative anteroposterior X-ray of left shoulders. (**A**) Reconstruction of the glenoid with BMA and a 15 degrees full wedge augmented central screw baseplate. The dotted blue line represents the native glenoid. There is concurrent plate fixation of a preoperative fatigue fracture of the spine of the scapula. (**B**) Reconstruction of the glenoid with BA and a 25-mm-long central peg baseplate.

**Figure 3 jcm-10-05274-f003:**
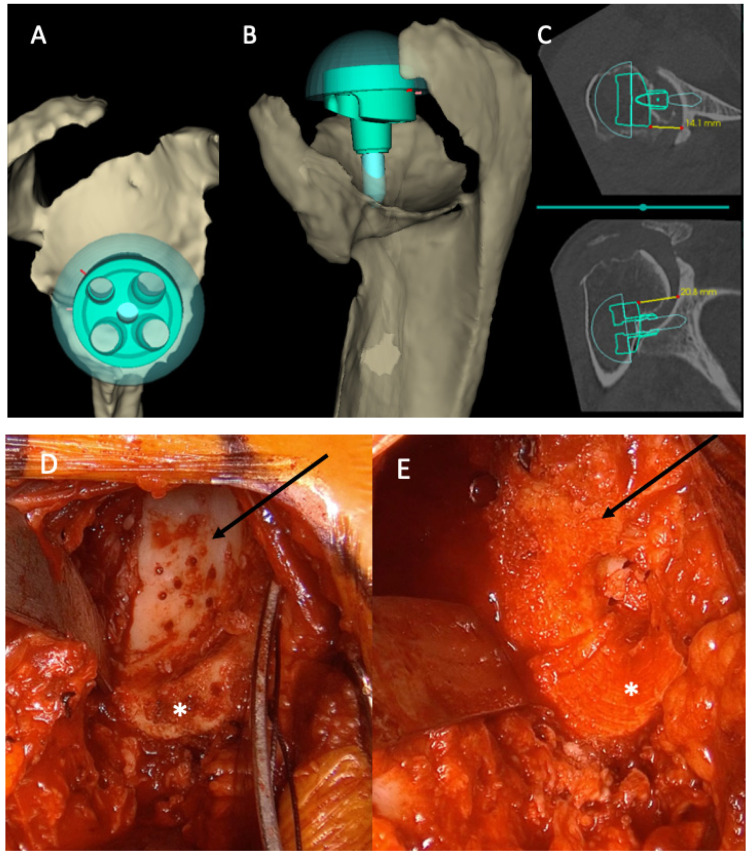
This technique compensates for massive bone defects and creates a more inferior tilt, transforming shear forces into compression ones. (**A**) Sagittal and (**B**) superior views of a 3-dimensional (3D) reconstruction of a right shoulder. Note the massive posterosuperior bone loss. (**C**) Planification reveals that metallic augmentation alone would not achieve optimal joint line restoration. (**D**) Intraoperative anterior view of the paleoglenoid (white asterix) and superior bone erosion (black line). (**E**) Glenoid reconstruction after humeral bone autograft (black arrow). The entire humeral head is hardly sufficient to compensate for the bony erosion. *: paleoglenoid. (**F**) Postoperative anteroposterior X-ray confirms that BMA allows for a large area of bony contact between the autograft (complete humeral head, dotted black line) and the native glenoid, correcting massive bone loss.

**Table 1 jcm-10-05274-t001:** Patient characteristics between the Bony-metallic augmentation (BMA) and Bony augmentation (BA) groups.

	Bony-Metallic Augmentation (*n* = 8 Patients)				Bony Augmentation (*n* = 8 Patients)				*p*-Value
	*N*	(%)			*N*	(%)			
	Mean	±SD	(Range)		Mean	±SD	(Range)		
Male sex	3	(37.5%)			5	(62.5%)			0.619
Operation on dominant side	7	(87.5%)			3	(37.5%)			0.119
Prior surgery	1	(12.5%)			0	(0.0%)			1.000
Tobacco use	1	(12.5%)			0	(0.0%)			1.000
Primary diagnosis									0.200
Primary OA	5	(62.5%)			8	(100.0%)			
Post-traumatic arthritis	1	(12.5%)			0	(0.0%)			
Dislocation arthropathy	1	(12.5%)			0	(0.0%)			
Revision	1	(12.5%)			0	(0.0%)			
Glenoid morphology									** <0.001 **
B1	1	(12.5%)			0	(0.0%)			
B2	0	(0.0%)			8	(100.0%)			
B3	1	(12.5%)			0	(0.0%)			
C	2	(25.0%)			0	(0.0%)			
D	2	(25.0%)			0	(0.0%)			
E3	2	(25.0%)			0	(0.0%)			
Age at index operation (yrs)	72.1	±11.7	(51.9	−83.9)	73.2	±6.8	(61.5	−84.1)	0.721
Body mass index	25.4	±4.1	(20.4	−31.6)	26.6	±3.0	(23.7	−31.0)	0.400
Weight (kg)	70.0	±11.1	(60.0	−85.0)	74.8	±11.2	(60.0	−95.0)	0.461
Height (cm)	166.6	±13.1	(140.0	−180.0)	167.5	±7.9	(155.0	−175.0)	0.635
Follow-up (months)	28.1	±15.0	(11.0	−51.0)	30.7	±10.8	(24.0	−55.4)	0.752
Inclination (°)	15.1	±12.0	(0.0	−34.0)	9.3	±7.5	(−7.0	−17.0)	0.528
Ante retroversion (°)	−22.5	±21.1	(−41.0	−12.0)	−26.1	±4.7	(−36.0	−−21.0)	0.494
Bone loss thickness (mm)	16.3	±3.8	(11.0	−21.0)	12.0	±0.0	(12.0	−12.0)	** 0.020 **

BMA—bony-metallic augmentation, BA—bony augmentation, OA—Osteoarthrosis. Underlined *p*-values indicate those below 0.05.

**Table 2 jcm-10-05274-t002:** Radiological data of each patient.

Patient	Glenoid Morphology	Inclination (°)	Ante/Retroversion (°)	Bone Loss Thickness (mm)
Patient 1	B2	10	−27	12
Patient 2	B2	9	−27	12
Patient 3	B2	−7	−21	12
Patient 4	B2	17	−36	12
Patient 5	B2	9	−23	12
Patient 6	B2	7	−23	12
Patient 7	B2	13	−28	12
Patient 8	B2	16	−24	12
Patient 9	D	5	12	11
Patient 10	E3	0	−36	14
Patient 11	B1	10	−10	12
Patient 12	C	16	−40	16
Patient 13	B3	27	−37	21
Patient 14	D	34	−41	20
Patient 15	E3	23	3	20
Patient 16	C	6	−31	16

**Table 3 jcm-10-05274-t003:** Pre- and postoperative data between the Bony-metallic augmentation (BMA) and Bony augmentation (BA) groups.

	Bony-Metallic Augmentation (*n* = 8 Patients)				Bony Augmentation (*n* = 8 Patients)				*p*-Value
	N	(%)			N	(%)			
	Mean	±SD	(Range)		Mean	±SD	(Range)		
Internal rotation									
preoperative									** **0.022** **
Thigh	5	(62.5%)			0	(0.0%)			
Buttock	3	(37.5%)			4	(50.0%)			
Sacrum	0	(0.0%)			3	(37.5%)			
Th12	0	(0.0%)			1	(12.5%)			
postoperative									0.220
Buttock	0	(0.0%)			1	(12.5%)			
Sacrum	0	(0.0%)			3	(37.5%)			
L5/L3	4	(50.0%)			3	(37.5%)			
Th12	2	(25.0%)			1	(12.5%)			
Th7	2	(25.0%)			0	(0.0%)			
*p*-value *	** 0.014 **				0.090				
Anterior forward flexion (°)									
preoperative	55.0	±38.5	(0.0	−95.0)	101.3	±31.8	(40.0	−150.0)	** 0.010 **
postoperative	141.3	±22.2	(90.0	−160.0)	145.0	±12.0	(130.0	−160.0)	0.915
improvement	86.3	±27.9	(55.0	−130.0)	43.8	±25.6	(10.0	−90.0)	** 0.013 **
*p*-value *	** 0.014 **				** 0.014 **				
Active external rotation (°)									
preoperative	14.4	±15.5	(0.0	−40.0)	10.6	±22.4	(−20.0	−45.0)	0.545
postoperative	29.4	±14.7	(15.0	−60.0)	49.4	±17.0	(30.0	−80.0)	**0.017**
improvement	15.0	±12.0	(−10.0	−30.0)	38.8	±16.2	(5.0	−60.0)	**0.009**
*p*-value *	** 0.027 **				** 0.014 **				
Constant score									
preoperative	18.8	±7.4	(6.0	−30.0)	34.5	±11.7	(12.0	−49.0)	** 0.013 **
postoperative	75.4	±10.4	(55.0	−87.0)	72.8	±14.8	(54.0	−91.0)	0.792
improvement	56.6	±10.1	(38.0	−72.0)	38.3	±16.7	(17.0	−69.0)	** 0.021 **
*p*-value *	** 0.014 **				** 0.008 **				
SSV									
preoperative	34.4	±21.9	(15.0	−70.0)	36.3	±14.1	(10.0	−50.0)	0.630
postoperative	83.8	±11.6	(65.0	−100.0)	85.5	±9.2	(70.0	−99.0)	0.915
improvement	49.4	±24.4	(10.0	−80.0)	49.3	±16.8	(35.0	−85.0)	0.958
*p*-value*	** 0.008 **				** 0.014 **				
ASES score									
preoperative	28.6	±14.1	(3.0	−47.0)	22.9	±11.8	(10.0	−35.0)	0.426
postoperative	76.8	±11.0	(57.0	−95.0)	89.1	±11.3	(67.0	−100.0)	** 0.045 **
improvement	48.1	±12.2	(31.0	−66.0)	66.3	±11.9	(45.0	−83.0)	** 0.027 **
*p*-value *	** 0.014 **				** 0.008 **				

* Between pre- and postoperative measurements; SSV, Subjective Shoulder Value; ASES, American Shoulder and Elbow surgeons. Underlined *p*-values indicate those below 0.05.

## Data Availability

All data relevant to the study are included in the article. Details regarding where data supporting reported results can be asked at the following e-mail address: hugo.bothorel@latour.ch.

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
