# Peer review of "Reverse Shoulder Arthroplasty with Bony and Metallic versus Standard Bony Reconstruction for Severe Glenoid Bone Loss. A Retrospective Comparative Cohort Study"

_jcm, 2021, doi:10.3390/jcm10225274_

Round 1
Reviewer 1 Report
the title of the article must be change: it is not a comparative study between the group BMA vs BA because as it is shown the 2 groups are not comparable:in term of pre op scales and of classification os glenoidal wear.
if you keep that title you suggest that it is a comparison of 2 treatments for 2 comparative cohorts and that you want too show that one treatment is better.
in that article you don't need too comparate the 2 cohorts: it is just the evaluation about 8 cases of reconstrtuction in cases of glenoidal wear in RSA. the clinical results are comparable . BUT if you want too show the early time results of BMA reconstructions or its safety you have to analyse that cases with a CT scan at revision for example; to see that maybe you have a better integration of the bio RSA with the augmented base plate.
In the results ; you have indeed one notch in BMA vs 0 in BA.
if you look the glenoidal wear: why use an inferior tilted glenoidal base plate when you have majority of posterior glenoidal Wear : Type B C and D.
I think the authors wanted to show the safety of the BMA technic but the design of the study " a comparisaon between two cohorts" in inappropriated.
maybe because of the many conflitcts of interest of the authors with the company?
Author Response
|
Reviewer #1's comments: |
|
|
|
Comments |
Answers/Corrections |
Line |
|
the title of the article must be changed: it is not a comparative study between the group BMA vs BA because as it is shown the 2 groups are not comparable: in term of pre op scales and of classification os glenoidal wear. if you keep that title, you suggest that it is a comparison of 2 treatments for 2 comparative cohorts and that you want to show that one treatment is better. |
Thank you for your comments. We understand your points and agree that the 2 groups of patients are not completely homogenous. However, the goal of the present study, as mentioned in the hypothesis, was to show “that the combined use of bone graft and metallic augments in severe glenoid bone loss during RSA is safe and effective”. The authors feel that this was best demonstrated by comparison to the current accepted treatment of reconstructing the glenoid (i.e. BA). Our intention is not to demonstrate that a treatment is better than the other one.
In addition, the similarities between the two groups permit, according to us, some degree of comparison.
First, there was no significant difference in the demographic data of patients in both groups. Patient characteristics showed no significant differences in terms of patient age, gender, BMI, tobacco usage, or affected side.
Second, there was also no significant difference in the pre-operative surgical variables such as history of prior surgery, primary diagnosis, glenoid inclination, or glenoid version.
Third, the surgical technique was the same, the two surgeons being trained by the same mentor.
Fourth, all prosthesis had been designed by the same the stems implant company and the stems were identical.
Consequently, the only relevant differences were 1) the baseplate and 2) the amount and the location of bone loss that imposed an additional type of augmentation in the BMA group.
Thus, even if not identical, we believe that these groups are comparable. We have stressed the heterogeneity of the groups in the limitations section. |
|
|
In that article you don't need to comparate the 2 cohorts: it is just the evaluation about 8 cases of reconstrtuction in cases of glenoidal wear in RSA. the clinical results are comparable. BUT if you want to show the early time results of BMA reconstructions or its safety you have to analyse that cases with a CT scan at revision for example; to see that maybe you have a better integration of the bio RSA with the augmented base plate. |
The decision of not only evaluating the results of BMA but also comparing it to BA was made, because we wanted to compare a previously non-described way of glenoid bone loss treatment (BMA) to a well-established and documented treatment (BA).
The authors agree that computed tomography (CT) is another modality for radiological comparison. This was not done for this group of patients and there are also some concerns that graft resorption may be underestimated with CT scans as a result of metallic artifacts of the baseplate and glenosphere (http://dx.doi.org/10.1016/j.otsr.2015.03.010).
Nevertheless, your comment has been taken into account and added in the Limitations section. |
Lines 289-291. |
|
In the results ; you have indeed one notch in BMA vs 0 in BA. |
The reported incidence of notching in the literature with RSA is up to 88%.(http://dx.doi.org/10.1302/0301-620X.93B9.25926) We reported a prevalence of 12.5% in the BMA group, of 0% in the BA group, the difference being not statistically significant.
We are aware that BMA do not prevent in all case scapular notching, mainly due to massive medialization due to glenoid erosion. Interestingly, it does not seem that BA does not prevent neither scapular notching compared with a traditional RSA.(https://doi.org/10.1016/j.jse.2017.07.020) |
|
|
If you look the glenoidal wear: why use an inferior tilted glenoidal base plate when you have majority of posterior glenoidal Wear : Type B C and D. |
You are perfectly right. We use a full wedge to transform shear forces into compressive forces. Our sentenced was thus unprecise. It has been change for “a 15 degrees full wedge augmented baseplate was screwed at the edge of the glenoid with the greatest bone loss”
|
Line 99. |
|
|
|
|
|
Maybe because of the many conflicts of interest of the authors with the company? |
The conflicts of interests have been clearly reported. However, it does not play a role. There is effectively many baseplates’ design like the one presented that could play a similar role.
The goal of the present article was not to push for a prosthesis but rather to describe an unreported elegant association to solve difficult surgical situations. Please refer to comment 1 of reviewer 2. |
|
Reviewer 2 Report
This is an interesting and well written comparative clinical study. In the current study the authors compare the outcome of the bone-metallic- vs. bone-augmentation of glenoid defects in RSA.
Unfortunately the sample size is very low, however, as this is a new concept without any previous publications in the literature. it shows that in certain situations a combination of bony and metallic glenoid augmentation appears to be feasible and safe.
I am convinced that these data are of great interest to your readers and recommend publication.
There are a few very minor points that need be specified / commented on:
- line 173: please explain AE (active elevation) as this may not be clear to all readers
- Figures 2 description: are you certain that the dotted blue line indicates the paleo glenoid, I'd rather call it "native" or the "neo-glenoid" but this is not an anatomic glenoid line
- Figures 3 description: please delete the duplicated text
Author Response
|
Reviewer #2's comments: |
|
|
|
Comments |
Answers/Corrections |
Line |
|
Materials and Methods |
||
|
This is an interesting and well written comparative clinical study. In the current study the authors compare the outcome of the bone-metallic- vs. bone-augmentation of glenoid defects in RSA. Unfortunately the sample size is very low, however, as this is a new concept without any previous publications in the literature. it shows that in certain situations a combination of bony and metallic glenoid augmentation appears to be feasible and safe. I am convinced that these data are of great interest to your readers and recommend publication. |
Thank you for your comment. Please see our answers below. |
|
|
Line 173: please explain AE (active elevation) as this may not be clear to all readers |
We deleted the AE from the text and replaced it with anterior forward flexion. |
Line 173, 174, 184, 196. |
|
Figures 2 description: are you certain that the dotted blue line indicates the paleo glenoid, I'd rather call it "native" or the "neo-glenoid" but this is not an anatomic glenoid line |
We corrected it.
The dotted line represents effectively the remnant glenoid. |
Line 116. |
|
Figures 3 description: please delete the duplicated text |
We corrected it. |
Lines 247 – 252. |